# Joint Evaluation of Fairness and Relevance in Recommender Systems with Pareto Frontier

## ABSTRACT

Fairness and relevance are two important aspects of recommender systems (RSs). Typically, they are evaluated either (i) separately by individual measures of fairness and relevance, or (ii) jointly using a single measure that accounts for fairness with respect to relevance. However, approach (i) often does not provide a reliable joint estimate of the goodness of the models, as it has two different best models: one for fairness and another for relevance. Approach (ii) is also problematic because these measures tend to be ad-hoc and do not relate well to traditional relevance measures, like NDCG. Motivated by this, we present a new approach for jointly evaluating fairness and relevance in RSs: distance from pareto frontier (DPFR). Given a user-item interaction dataset, we compute their Pareto frontier for a pair of existing relevance and fairness measures, and then use the distance from the frontier as a measure of the jointly achievable fairness and relevance. Our approach is modular and intuitive as it can be computed with existing measures. Experiments with 4 RS models, 3 re-ranking strategies, and 6 datasets show that the existing metrics have inconsistent associations with our Pareto-optimal solution, making DPFR a more robust and theoretically well-founded joint measure for assessing both fairness and relevance.

## KEYWORDS

evaluation, relevance, fairness, pareto frontier, recommendation

**ACM Reference Format:**
Anonymous Author(s). 2024. Joint Evaluation of Fairness and Relevance in Recommender Systems with Pareto Frontier. In *Proceedings of Conference Title (Conference acronym 'XX).* ACM, New York, NY, USA, 17 pages. https://doi.org/XXXXXXX.XXXXXXX

## 1 INTRODUCTION

Relevance and fairness are important aspects of recommender systems (RSs). Relevance is typically evaluated using well-known ranking measures (e.g., NDCG), while various fairness measures for RSs exist [1, 47]. Some fairness measures integrate relevance, so that they evaluate fairness w.r.t. relevance. The problem with these joint measures is that they tend to be ad-hoc, unstable, and they do not account very well for both aspects simultaneously [37]. Another way of evaluating relevance and fairness is to use a different measure for each aspect. However, this does not always provide a reliable joint estimate of the goodness of the models, as it may

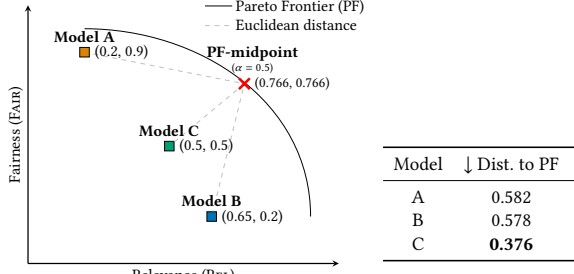

**Figure 1:** $(x, y)$ **denotes the pair of relevance and fairness score. Example: Model A is best for fairness, Model B is best for relevance, and Model C is the closest to the Pareto Frontier (PF) midpoint, when relevance and fairness are equally weighted ($\alpha = 0.5$). Averaging relevance and fairness (Avg) leads to falsely concluding that Model A is best for both aspects. Note that distance to PF also beats other existing measures of fairness and relevance (see §5.4).**

have two different best models: one for fairness and another for relevance. This can be avoided by aggregating the scores of the two measures into a single score, or by aggregating the resulting model rankings into one using ranking fusion. These approaches are also problematic because: (i) the scores of the two measures may have different distributions and different scales, making them hard to combine; (ii) the two measures may not even be computed with the same input, making their combination hard to interpret (relevance scores are computed for individual users and then averaged, while fairness measures for individual items are typically based on individual item recommendation frequency); and (iii) the resulting scores are less understandable as it is unknown how close the models are to an ideal balance of fairness and relevance.

To address the above limitations, we contribute an approach that builds on the set of all Pareto-optimal solutions [6]. Our approach addresses issue (i) and (ii) above by avoiding direct combination of measures. We directly address (iii) by computing the distance of the model scores to a desired fairness-relevance balance. Our approach uses Pareto-optimality, a popular concept in multi-objective optimization problems across domains, including RSs [39]. A recommendation is Pareto-optimal if there are no other possible recommendations with the same REL score that achieve better fairness.[1] In other words, given Pareto-optimal solutions, we cannot get other recommendations that empirically perform better, unless relevance is sacrificed. In our approach we combine existing FAIR measures and REL measures as follows. We build a PF that first maximises relevance, finds the best fairness achievable under

---
[1] The opposite is also true, but in RS scenario the REL score is usually the primary objective, not the FAIR score.

the relevance constraint, and then jointly quantifies fairness and relevance as the distance from an optimal solution, see Fig. 1.

Our approach, *Distance to PF of Fairness and Relevance* (DPFR) has several strengths. First, DPFR is *modular*; it can be used with well-known existing measures of relevance and fairness. DPFR is also *tractable* as one can control the weight ($\alpha$) of fairness w.r.t. relevance. As the resulting score is the distance to the scores of a traditional relevance measure and a well-known fairness measure, DPFR is also *intuitive* in its interpretation. Most importantly, DPFR is a principled way of jointly evaluating relevance and fairness based on an empirical best solution that uses Pareto-optimality. Experiments with different RS models, re-ranking approaches and datasets show that there exists a noticeable gap between using current measures of relevance and fairness and our Pareto-optimal joint evaluation of relevance and fairness. This gap is bigger in larger datasets and when using rank-based relevance measures (i.e., MAP, NDCG), as opposed to set-based relevance measures (i.e., Precision, Recall).

In this work, we focus on **individual item fairness**. This type of fairness is commonly defined as all items having equal exposure, where exposure typically refers to the frequency of item appearance in the recommendation list across all users [24, 34, 36]. Individual item fairness is important in ensuring that each item/product in the system has a chance to be recommended to any user [17].

## 2 RELATED WORK

Evaluating fairness and relevance together is a type of multi-aspect evaluation. However, none of the existing multi-aspect evaluation methods [22, 23, 32] can be used in this case as these methods require separate labels that are unavailable in RS scenarios. Specifically, it is not possible to label an item as 'fair', because item fairness depends on other recommended items. The same item can be a fair recommendation in one ranking, but unfair in another. In RSs, fairness is typically defined as treating users or items without discrimination [3]. This is often quantified as the opportunity for having equal relevance (for users) or exposure (for items) [3, 46], computed either individually or for groups of items/users [35, 53].

The problem of evaluating RS relevance and fairness together is further aggravated by the fact that improved fairness is often achieved at the expense of relevance to users [26]. We posit that this trade-off makes multi-objective optimization a suitable solution. Pareto optimality is a well-known objective for such optimization, and it has been previously used in RS but only to recommend items to users [10, 39, 49, 55]. Because the true PF is often unknown due to the problem complexity [2, 16], previous work has used the model's training loss w.r.t. two different aspects [20] or scores from different models [30, 33] to generate the PF. Our work differs from this prior work in terms of both the purpose of using Pareto-optimal solutions, and the nature of the PF. Specifically, we exploit Pareto-optimality through PF as a robust *evaluation* method, instead of as a recommendation method. In addition, our generated PF is based on the ground truth (i.e., the test set), a common RS evaluation approach, instead of the recommender models' empirical performance, which may not be optimal. Thus, our PF is also model-agnostic, as opposed to the PF in [49]. Our approach differs also from FAIR [9] since the PF considers the empirically achievable optimal solution based on the dataset, while FAIR compares against the desired fairness

distribution which might not be achievable. Lastly, the approach in [33] selects the optimal solution based on its distance to the utopia point (the theoretical ideal scores), whereas the utopia point may not be realistic due to dataset or measure characteristics [28, 36]. Since our PF is generated based on test data, any of its solutions is empirically achievable.

## 3 DISTANCE TO PARETO FRONTIER (DPFR)

We present definitions (§3.1), and then explain DPFR in different steps: given a Fair and a Rel measure, how to generate PF based on the ground truth data in the test set (§3.2); how to choose a reference point in the PF based on $\alpha$ (e.g., the midpoint for $\alpha = 0.5$) (§3.3); and how to compute the distance of the Fair and Rel scores to the reference point with a distance measure $d$ (§3.3). Additionally, we present a computationally efficient adaptation of DPFR (§3.4).

### 3.1 Definitions

We adapt the Pareto-optimality definition [43] to our case. Here, the multi-objective problem is finding the optimum Fair score $s_f$, and Rel score, $s_r$ from a list of possible recommendations across all users. We define the tuple $s = (s_r, s_f) \in S$, where $S$ is the Cartesian product of all possible Rel and Fair scores. The relation $\geq_A$ means 'better or equal to' according to an aspect $A \in \{\text{Rel}, \text{Fair}\}$. The relation $>_A$ is defined similarly.

**Def. 1** (Pareto Dominance). A tuple $s = (s_r, s_f)$ dominates $s' = (s'_r, s'_f)$ iff $s$ is partially better than $s'$, i.e., $s_r \geq_{\text{Rel}} s'_r$ and $s_f \geq_{\text{Fair}} s'_f$, in addition to $s_r >_{\text{Rel}} s'_r$ or $s_f >_{\text{Fair}} s'_f$.

**Def. 2** (Pareto Optimality). A solution (recommendation list) that has Rel and Fair scores of $x = (x_r, x_f) \in S$ is Pareto-optimal iff there is no other solution with $x' = (x'_r, x'_f) \in S$ that dominates $x$.

**Def. 3** (Pareto Frontier). The set of all Pareto-optimal tuples.

### 3.2 Pareto Frontier generation

Given user-item preference data (e.g., test set), the aim is to explore the empirical, maximum feasible fairness towards individual items considering all items in the recommendation, that satisfies Pareto-optimality w.r.t. fairness and an average relevance score across users, e.g., MAP@10 = 0.9.[2] This is done to measure how far a model performance is, from these Pareto-optimal solutions. Enumerating all possible recommendations for users and items to find the complete set of Pareto-optimal solutions is computationally infeasible, and there is no analytical solution either. Instead, we contribute an algorithm that iteratively builds upon a maximally relevant initial recommendation list. Our algorithm iteratively finds Pareto-optimal recommendations by prioritising relevance over fairness, as recommendations are usually optimised for relevance (with or without fairness). This prioritisation is known as lexicographic optimization [40]. We call our algorithm Oracle2Fair (full technical description in App. B). Our algorithm generates the PF of fairness and relevance in two steps: **(1) initialisation** of the recommendations with an *Oracle* (App.B, Algorithm 1). The *Oracle* generates a recommendation with the highest empirical score for relevance, based on user interactions that are part of the test set.

---

[2]This is how Fair and Rel measures are usually computed.

This step is followed by **(2) replacements** to make the recommendations as *Fair* as possible; at the end of this algorithm, the FAIR scores should reach the empirically fairest score while maintaining as much relevance as possible. Throughout the PF generation, items in a user's train/val split are not recommended to the same user. Henceforth, *relevant items* refers to the items in a user's test split.
**(1) Initialisation**. The Oracle recommends at most $k = 10$ relevant items, from the $n$ items in the dataset, to each of the $m$ users in the test split, one user at a time. The recommendation begins with users having exactly $k$ items in the test split, as only these items can be recommended to those users to gain the maximum relevance. For other users, the recommendations are made maximally relevant and fair as follows: if a user has more than $k$ relevant items, we pick $k$ items with the least exposure among them. Item exposure is computed based on what has been recommended to other users who already have exactly $k$ items. Note that this process is not trivial (see App. B, Algo 1, ll. 2–17). If a user has less than $k$ relevant items, we recommend those items at the top (to maximise top-weighted REL measures) and fill the rest of their recommendation slots with the least exposed items in the dataset (Algo 1, ll. 18–36). This least-exposure prioritisation strategy ensures that the solutions are Pareto-optimal.
**(2) Replacements**. The algorithm iteratively replaces the recommended items to achieve maximum fairness, such that each replacement results in a fairer recommendation than the previous. We compute the FAIR and REL measures after each replacement as follows. The most popular item, which is recommended most often, is replaced with one of these item types, in decreasing order of priority: an unexposed item, then the least popular item in the recommendation; this increases fairness from the previous recommendations. We do this one item and one user at a time, starting with the users that have the most popular item at the bottom of their recommendation list, to ensure that the decrease in relevance is minimum as the replacement item is mostly not relevant to that user. Nonetheless, the ORACLE2FAIR prioritises replacing the recommendations of users for whom the replacement item is relevant (if any). As fairness increases and relevance decreases/stays the same from the previous recommendation, the new recommendation is also Pareto-optimal. We continue the replacement until the maximum times any item is recommended is $\lceil km/n \rceil$, i.e., the upper bound of how many times an item can be recommended, if all items in the dataset must appear in the recommendation as uniformly as possible. To ensure maximum REL scores (especially in top-weighted measures), each time a replacement takes place, we rerank the recommendations based on descending relevance.

The resulting pairs of (REL, FAIR) scores corresponding to Pareto-optimal recommendations from this process make up the PF. If there are duplicates in the REL value, we only keep the best FAIR score for a single value of REL. While it cannot be verified in reasonable time that the resulting PF exactly matches the theoretical PF, this is one of the closest ways to build the full PF, as opposed to building the PF from trained models scores (§2).

### 3.3 Distance computation

For each pair of FAIR and REL measures, we find a reference point using a tunable parameter $\alpha \in [0, 1]$; $\alpha = 0$ means only relevance

is accounted for, and $\alpha = 1$ means only fairness is accounted for. Next, we explain how to compute the reference point. We first use the following equation to find the length of a subset $T$ of the PF: $lenPF(T) = \sum_{t=1}^{|T|-1} d_E(x^t, x^{t+1})$. Given that $P$ is the set of all Pareto-optimal solutions, $x^t = (x_r^t, x_f^t)$ is the pair of Pareto-optimal solutions $(x_r, x_f)$ with the $t$-th highest $x_r$ in $P$, and $d_E$ is the Euclidean distance. The overall PF length is $lenPF(P)$ or simply $lenPF$.

The reference point is $s_\alpha = x^{t'}$, where $t'$ is computed as follows: $t' = \arg\min_{j \in [1,...,|P|-1]} \left| lenPF(T^j) - \alpha \cdot lenPF \right|$, where $T^j$ is a subset of $P$ containing the $j$ highest $x_r$ scores. In other words, the reference point is a point in the PF whose cumulative traversal distance is the closest to the $\alpha$-weighted PF distance travelled from the first point in the PF. Essentially, the reference point $s_\alpha$ is how far the PF is traversed, from the pair with the best REL score to the one with the best FAIR score, multiplied by $\alpha$. As the PFs may have different density of points along the frontiers, the reference point is not computed based on a percentile (e.g., median) to avoid bias towards the denser part. Next, the distance between each model's $(x_r, x_f)$ scores and the reference point $s_\alpha$ is computed with a distance measure $d$ that accommodates 2d-vectors. The model with the closest distance is the best model in terms of both relevance and fairness, given the weight $\alpha$. We call this *Distance to Pareto frontier of Fairness and Relevance* (DPFR).

### 3.4 Efficient computation of Pareto Frontier

Generating the PF as in §3.2 is computationally expensive. An efficient alternative is to compute a subset of the PF. We pick a fixed amount of Pareto-optimal solutions to compute, $p$ (e.g, 10). However, to reliably approximate the PF, these solutions should be spread according to the PF distribution, as opposed to e.g. only computing the first $p$ points of the PF. The spread of the points is important, as the reference point in DPFR is computed based on the overall estimated PF. In the estimated PF, the first point corresponds to the measure scores of the initial recommendation given by the Oracle, and the rest are spread evenly throughout the PF generation. To select at which point of the ORACLE2FAIR algorithm the measures should be computed, we first estimate the total number of replacements needed by examining the distribution of recommended items frequency. This is done by getting the individual frequency count of all items in the recommendation, and subtracting the ideal upper bound of item count $\lceil km/n \rceil$ (§3.2) from each count. The number of expected replacements is computed as the sum of the difference between the item frequency count and the ideal upper bound of item count in §3.2. Items with recommendation frequency counts less than the upper bound are excluded from the summation. With the estimated total number of replacement $numRep$, we set to compute the measures every $numRep \, \text{div} \, (p - 1)$ replacements done by ORACLE2FAIR, such that the measures are computed a total of $p - 1$ times + 1 time before the replacement starts. These $p$ points are spread evenly in terms of distance in the PF, which is important as DPFR is a distance-based measurement.

### 4 EXPERIMENTAL SETUP

We study how our joint evaluation approach, DPFR, compares to existing single- and multi-aspect evaluation measures of relevance

**Table 1: Statistics of the preprocessed datasets.**

| dataset | #users ($m$) | #items ($n$) | #interactions | sparsity (%) |
|---|---|---|---|---|
| Lastfm [5] | 1,859 | 2,823 | 71,355 | 98.64% |
| Amazon-lb [29] | 1,054 | 791 | 12,397 | 98.51% |
| QK-video [50] | 4,656 | 6,423 | 51,777 | 99.83% |
| Jester [12] | 63,724 | 100 | 2,150,060 | 66.26% |
| ML-10M [13] | 49,378 | 9,821 | 5,362,685 | 98.89% |
| ML-20M [13] | 89,917 | 16,404 | 10,588,141 | 99.28% |

and fairness. Next, we present our experimental setup.[3] The experiments are run in a cluster of CPUs and GPUs (e.g., Intel(R) Xeon(R) Silver 4214R CPU @ 2.40GHz, AMD EPYC 7413 and 7443, Titan X/Xp/V, Titan RTX, Quadro RTX 6000, A40, A100, and H100).

**Datasets.** We use six real-world datasets of various sizes and domains: e-commerce (Amazon Luxury Beauty, i.e., Amazon-lb [29]), movies (ML-10M and ML-20M [13]), music (Lastfm [5], videos (QK-video [50]), and jokes (Jester [12]). The datasets are as provided by [54], except for QK-video, which we obtain from [50]. Statistics of the datasets are in Tab. 1 and extended statistics are in App. A.

**Preprocessing.** We remove duplicate interactions (we keep the most recent). We keep only users and items with $\geq 5$ interactions. We convert ratings equal/above the following threshold to 1 and discard the rest: for Amazon-lb and ML-*, the threshold is 3, as their ratings range between $[1, 5]$ and $[0.5, 5]$ resp.; the threshold for Jester is 0, as the ratings range in $[-10, 10]$. Lastfm and QK-video have no ratings. QK-video has several interaction types, and we only use the 'sharing' interactions. For Jester, we remove users with $> 80$ interactions, to provide a large enough number of item candidates for recommendation during testing.[4]

**Data splits.** To obtain the train/val/test sets for Amazon-lb and ML-*, we use global temporal splits [27] with a ratio of 6:2:2 on the preprocessed datasets. Global random splits with the same ratio are used for the other datasets that have no timestamps. From all splits, we remove users with $< 5$ interactions in the train set.

**Measures.** We measure relevance (REL) with Hit Rate (HR), MRR, Precision (P), Recall (R), MAP, and NDCG. We focus on individual item fairness (FAIR), and measure it, as per [36], with Jain Index (Jain) [14, 56], Qualification Fairness (QF) [56], Gini Index (Gini) [11, 24], Fraction of Satisfied Items (FSat) [34], and Entropy (Ent) [34, 42]. We also use joint measures (FAIR+REL): Item Better-Off (IBO) [41],[5] Mean Max Envy (MME) [41], Inequity of Amortized Attention (IAA) [3, 4], Individual-user-to-individual-item fairness (II-F) [8, 48], and All-users-to-individual-item fairness (AI-F) [8]. We denote by ↑/↓ measures where higher/lower is better. DPFR is computed with Euclidean distance and $\alpha = 0.5$ (PF midpoint). For all runs, we compute the results at $k = 10$.

**Recommenders.** We select 4 well-known collaborative filtering-based recommenders: item-based $K$-nearest neighbour (ItemKNN) [7], Bayesian Personalised Ranking (BPR) [38], Variational Autoencoder with multinomial likelihood (MultiVAE) [19], and Neighbourhood-enriched Contrastive Learning (NCL) [21], with RecBole's

---

[3]Our code will be made public upon acceptance.

[4]Some users in Jester have interacted with almost all 100 items. If a user has 80 items in the train/val set, there would only be 20 candidate items to recommend during test, which makes it easier to achieve higher relevance.

[5]The measure Item Worse-Off is not used as its formulation is highly similar to IBO.

---

implementation and hyperparameter tuning [54]. We train for 300 epochs with early stopping, and keep the configuration with the best NDCG@10 during validation. Each user's train/val items are excluded from their recommendations during testing.

**Fair Re-rankers.** To have fairer recommendations, we reorder the top $k'$ items that are pre-optimised for relevance. Ideally $k' > k$ to allow exposing items that are not in the top $k$. As there are very few relevant items per user in RS datasets,[6] $k'$ should not be too big (e.g., 100). So, we re-rank the top $k' = 25$ items for each dataset and each model using three methods: GS, CM, and BC (explained below). The re-ranking is done separately per user for CM and BC, or altogether for GS, when considering all $k'm$ recommended items, where $m$ is the number of users. Other fair ranking methods exist, but we do not use them as they apply to group or two-sided fairness only (e.g., [34, 51, 52]), or to stochastic rankings only (e.g., [31, 48]), or do not scale to larger datasets (e.g., [3, 41]).

*1. Greedy Substitution (GS) [46]* is a re-ranker for individual item fairness. We modify the GS algorithm, to replace the most popular items with the least popular ones, both considering how many times an item is at the top $k'$ recommendations for all users (App. C). As such, items can be swapped across users. To determine which items are most popular (i.e. to be replaced) and least popular (replacement items), the parameter $\beta = 0.05$ is used. We pair these two item types, and for each pair, we calculate the loss of (predicted) relevance if the items are swapped. We then replace up to 25% of the initial recommendations, starting from item pairs with the least loss.

*2. COMBMNZ (CM) [18]* is a common rank fusion method. Two rankings are fused for each user: one based on the (min-max) normalised predicted relevance score and another based on the coverage of each top $k'$ item (to approximate fairness). We calculate item coverage only based on their appearance in the top $k$ across all users and min-max normalise the score across all users. As favouring items with higher coverage would boost unfairness, we generate the ranking using 1 minus the normalised coverage. CM uses a multiplier based on the item appearance count in the two rankings above; this count is also only based on the top $k$. The resulting ranking is a fused ranking of fairness and relevance.

*3. Borda Count (BC)* is a common rank fusion method. For each user, we combine the original recommendation list and the rankings based on increasing item coverage, as in CM. Unlike CM, BC uses points. Higher points are given to items placed at the top. The result is a fused ranking of fairness and relevance.

## 5 EXPERIMENTAL RESULTS

We now present the evaluation scores of 16 runs (4 recommenders x 3 re-rankers, including no reranking) (§5.1). The relevance and fairness scores of these runs are the input to our DPFR approach. Not all combinations of evaluation measures are suited for PF. We explain this in §5.2. We present the generated PF (§5.3) and compare existing measures to DPFR (§5.4). We compare the results of efficient DPFR to other joint evaluation approaches (§5.5).

### 5.1 Groundwork runs

The scores of REL, FAIR, and FAIR+REL measures for our 16 runs are shown in the appendix (Tab. 6–7). Two main findings emerge

---

[6]The median number of relevant items per user across all datasets is 2–53, see App. A,

from Tab. 6–7. First, for all six datasets, **none of the best models according to Rel are also the best according to Fair measures**. This is similar to our toy example (Fig. 1), where one model ranks highest for fairness and another for relevance. Second, **the five Fair+Rel measures have no unanimous agreement on the best model**. IBO has a different best model from the others in 4/6 times, but sometimes agrees with one or more Fair measures. MME and AI-F agree on the best model 5/6 times, and sometimes agree on the best model with Fair measures. The best model according to IAA and II-F is always the same, and 4/6 times the same as the best model based on the Rel measures. The overall picture is inconclusive, with some Fair+Rel measures aligning more with Fair measures, and others aligning more with Rel measures.

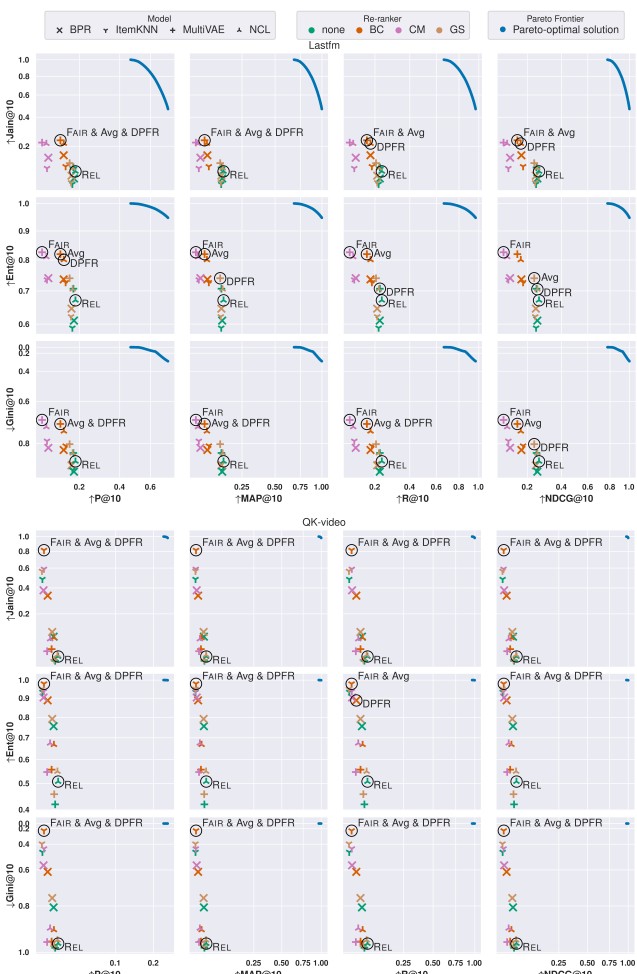

**Figure 2: Pareto Frontier of fairness and relevance (in blue) and recommender scores for Lastfm and QK-video on exponential-like scales. Rel, Fair, Avg (mean of Rel, Fair), and DPFR are the best model per evaluation approach.**

## 5.2 Measure compatibility with DPFR

Which pairs of Rel and Fair measures are suitable to generate the PF? We answer this based on the PF slope. The slope is calculated using the two endpoints of the PF, i.e., the start and end of the Oracle2Fair algorithm. A slope of zero means the Rel scores of the PF vary, but the Fair scores do not. As we compute the PF for multiple measures simultaneously, we expect a zero gradient for cases where the initial recommendation according to a Fair measure is already the fairest, even if other Fair scores are not. An undefined gradient value occurs when the initial recommendation is already the fairest and the most relevant according to a pair of Fair and Rel measures. Thus, we posit that a PF with a gradient value other than zero or undefined makes the corresponding pair of measures fit for PF generation (it allows for trade-offs in both aspects). The Rel-Fair measure pairs that are fit for DPFR based on their gradient are: {P, MAP, R, NDCG} × {Jain, Ent, Gini}. Only results from these pairs are shown henceforth. Next, we explain what causes an undefined or zero gradient for some measures.

**Causes of zero/undefined gradient.** Generating the PF requires a ranking of items. Any score that is based on a single relevant item, e.g., HR and MRR, is by design not suitable. Out of the Fair measures, QF and FSat sometimes behave inconsistently depending on the dataset properties, as follows. A dataset with relatively few relevant items can already be made maximally fair at the start of the PF generation, as QF quantifies fairness with ignorance to frequency of exposure; the score does not change as long as the same set of items appears in the top $k$ recommendations of all users, no matter how many times each. When all items in the dataset already occur in the initial recommendations of our Oracle, nothing can be done to improve QF. For FSat, in few cases, the score is already maximum at the start of the PF generation. A maximum FSat score is achieved when all items in the dataset have at least the maximum possible exposure, if the available recommendation slots are shared equally across all items.[7] In principle, QF and FSat can still be used for DPFR when the initial recommendation by Oracle is not the maximum yet. Otherwise, the interpretation would be less meaningful in joint evaluation, as there is no trade-off between different aspects.

## 5.3 The generated PF

Fig. 2 shows the PF plots of the pairs of Fair and Rel measures that are suited for DPFR, only for Lastfm and QK-video, which are representative of the overall trends in all our datasets (see Fig. 4 in the Appendix). The scores plotted are those computed in §3.2. The corresponding scores of our recommendation models are in App. D. We see that, as the recommendations are made fairer, the generated PF for all datasets is a series of monotonic scores of Fair, specifically monotonic increasing Fair scores (except ↓Gini), and the remaining measures are monotonic decreasing. The monotonic property theoretically and empirically holds for the Fair measures, as we replace an item with the most exposure by another item with the least exposure, thereby making the recommendation fairer. Note that some users do not have exactly $k$ items in the test set, so the perfect relevance score cannot be reached for Precision@$k$ and

---

[7]$\lfloor km/n \rfloor$ times (the total number of recommendation slots across users divided by the number of items).

Recall@$k$ [28]. NDCG and MAP are implemented with normalisation[8] so that they can still achieve a score of 1 in this situation.

The datasets which were randomly split as they have no timestamps (QK-video, Jester) have relatively short, compact PF. This happens because the random split results in a uniform distribution of items in each split, which means that items in the test split are quite diverse (64–100% of all unique items in the dataset). Considering that the Oracle2Fair algorithm starts by recommending items in the test split and stops when the recommendation reaches the fairest, there is not much room for change in Fair scores, as the initial recommendation is already rather fair. Additionally, there are not many relevant items per user in these datasets (i.e., the median for both datasets is 6 or less); random non-relevant items were chosen to make up for the remaining recommendations.[9] Thus, the PF generation decreases relevance only marginally in 2/6 cases. Correspondingly, we find that in QK-video and Jester, there exist Pareto-optimal recommendations, that are close to maximally fair and maximally relevant, with the exception of P@10. These can be seen in the measure pairs of {MAP, R, NDCG} × {Jain, Ent, Gini}, where the PF is close to the coordinates of $(1, 1)$, or $(0, 1)$ for the measure pairs with Gini. Thus, in theory, a fair recommendation does not necessarily have to sacrifice relevance.

## 5.4 Agreement between measures

We study the agreement between DPFR and other evaluation approaches in ranking our 16 runs from best to worst. Low agreement means that the other approaches have few ties to the Pareto-optimal solutions that DPFR uses, and vice versa. We compare DPFR to (a) existing Rel and Fair measures, (b) existing joint Fair+Rel measures (§4), and also (c) the average (arithmetic mean) of Fair and Rel scores from the selected measure pairs that are used to generate the PFs. To compute the average for a measure where lower values are better (i.e., Gini), we compute 1−the Gini score instead.

*5.4.1 Comparison of existing measures to DPFR.* We find that for all datasets and all measure pairs, **the best model as per DPFR is always different from the best model as per Rel measures**. Moreover, **half the time, the best model as per DPFR is different from the best model as per a Fair measure**. Existing Fair+Rel measures tend to have the same best model as either Fair or Rel measures (73.3% of the time), instead of having a more balanced evaluation of both aspects. These findings are expected as existing joint evaluation measures use relevance in their formulation differently than the Rel measures. Overall, the best model found with DPFR is less skewed towards relevance or fairness.

*5.4.2 Correlation of measures.* For each dataset, we compute the Kendall's[10] $\tau$ [15] correlations between the ranking given by DPFR and by the joint evaluation baselines (see Fig. 3). Rankings are considered equivalent if $\tau \geq 0.9$ [23, 44]. We see similar agreement trends in datasets where recommenders have higher Rel scores (Lastfm and Jester) or lower (Amazon-lb and QK-video). Overall, most times DPFR orders models differently ($\tau < 0.9$) than all

---

[8]Only the first min ($|R_u^*|, k$) items in a user $u$'s recommendations are considered, where $R_u^*$ is the set of relevant items for user $u$.

[9]The randomly-split Lastfm does not have a short PF because on average it has more relevant items per user compared to QK-video and Jester (see App. A).

[10]Ties are handled, unlike in Spearman's $\rho$.

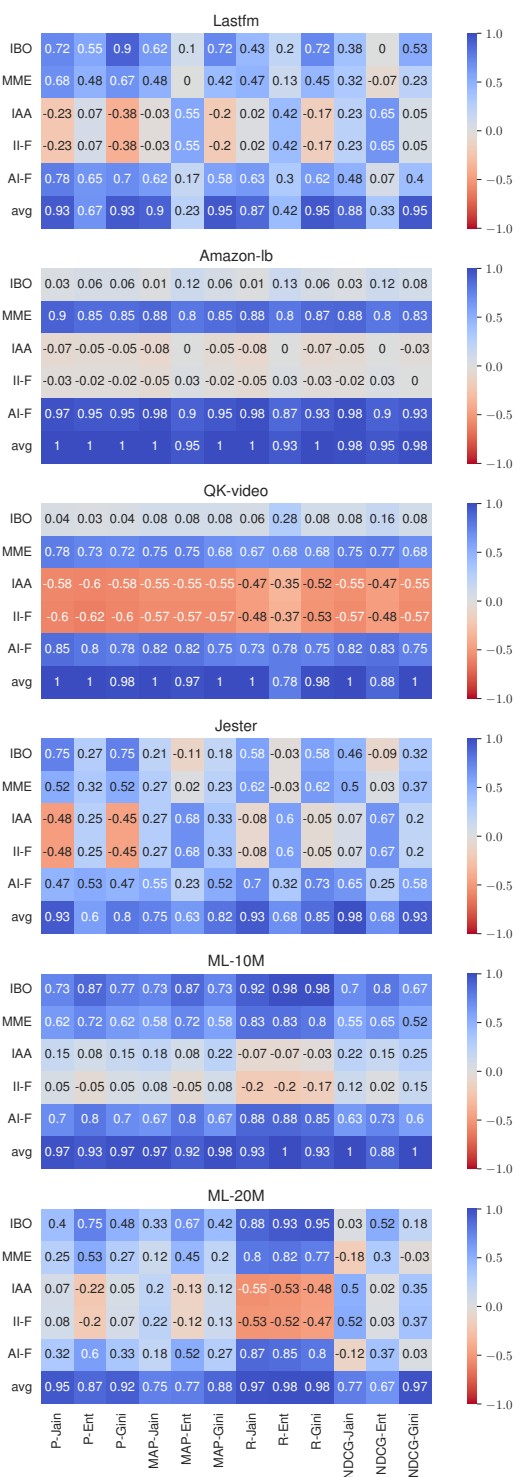

**Figure 3: Kendall's $\tau$ correlation heatmap between the rank ordering of existing joint evaluation measures (including the average of Fair and Rel scores, avg), and DPFR.**

Fair+Rel measures except AI-F. We see similar trends between IAA and II-F, and between MME and AI-F. IBO can have similar trends as MME (except for Amazon-lb and QK-video). For all datasets, IAA and II-F have overall either weak or negative $\tau$ with DPFR (e.g., $[-0.2, 0.25]$ for ML-10M and $[-0.62, -0.35]$ for QK-video). A notable exception is DPFR with {MAP, R, NDCG} × Ent for Lastfm and Jester, where we see moderate correlations, $\tau \in [0.42, 0.68]$.

Ent differs from this trend because DPFR with {MAP, R, NDCG} × {Jain, Gini} has PF gradients of greater magnitude. This only affects Lastfm and Jester (they have higher Rel scores than the other datasets). DPFR with P has different patterns from other Rel measures: the raw DPFR scores of pairs involving P are lower on average, as the scores from Oracle do not start from 1, but much less, and therefore closer to the models' scores (Fig. 2).) Meanwhile, IBO has varying $\tau$ across datasets: a huge range of $\tau$, i.e., $[0.00, 0.9]$ for Lastfm, weak correlations $[0.01, 0.13]$ for Amazon-lb, and moderate to strong correlations $[0.67, 0.98]$ for ML-10M. These variations might be because IBO is based on the number of items satisfying a certain criterion, rather than an average of scores across users and/or items, i.e., how other Fair+Rel measures are defined.

Among the joint measures, AI-F correlates the strongest with DPFR, as both AI-F and DPFR, indirectly or directly, consider the recommendation frequency of each item and compare it with that of other items. However, the rank orderings given by AI-F are not equivalent to DPFR, as $\tau < 0.9$ for 5/6 datasets (excl. Amazon-lb). For the same measure-pair and between datasets, the $\tau$ of AI-F and DPFR also varies a lot. E.g., $\tau = 0.07$ for NDCG-Ent for Lastfm, but $\tau = 0.9$ for Amazon-lb. We thereby do not recommend using any of the Fair+Rel measures ( none correlates with Pareto optimality).

Taking the mean of Fair and Rel scores (avg) at a glance seems to correlate highly with DPFR. However, while it gives equivalent rankings ($\tau \geq 0.9$) in some cases (e.g., for Amazon-lb, most of ML-10M and QK-video, and half of ML-20M), it only does so for (1) datasets with lower Rel scores (Lastfm, QK-video), i.e., in cases where all models perform poorly, we have low variance in Rel, which leads to fairness dominating both avg and DPFR; (2) datasets with low variance in Fair scores (ML-*). In such cases, quantifying the evaluation jointly is challenging as one aspect dominates over the other. In the other datasets, the rank ordering given by the average is inconsistent: sometimes $\tau \geq 0.9$ for one dataset, but not for the others. This inconsistency between datasets holds for all measure pairs, except for P-Jain and NDCG-Gini. Due to these inconsistencies, we discourage using the arithmetic mean.

Overall, our correlation analysis shows that existing joint Fair+Rel evaluation measures cannot be used as a reliable proxy for DPFR.

*5.4.3  Best model disagreement.* We take a closer look at how DPFR relates to computing averages, as they are similar approaches in terms of combining scores from a measure pair. As comparing the raw scores of DPFR and the average is invalid, we instead count the disagreement between the best model based on DPFR and the mean of Fair and Rel scores (Tab. 2). The aim is to study whether one would come to the same conclusion regarding the best model, using the two different joint evaluation approaches.

Among the 12 measure pairs that are fit for DPFR, we find that **the best model according to DPFR is not always the same according to the average of Fair and Rel scores of the same**

**Table 2: The percentage of best model disagreement when taking the mean of Fair and Rel scores as opposed to using DPFR, separated by the Rel measure type. P@$k$ and R@$k$ are set-based, NDCG and MAP are rank-based. We only consider the 12 measure pairs with a nonzero, defined gradient (§5.2).**

|  | Set-based | Rank-based | All |
|---|---|---|---|
| Lastfm | 50.00 | 66.67 | 58.33 |
| Amazon-lb | 0.00 | 0.00 | 0.00 |
| QK-video | 16.67 | 0.00 | 8.33 |
| Jester | 16.67 | 83.33 | 50.00 |
| ML-10M | 0.00 | 66.67 | 33.33 |
| ML-20M | 0.00 | 50.00 | 25.00 |
| All datasets | 13.89 | 44.44 | 29.17 |

**pair**; in one case the disagreement is up to 58% of the time (i.e., for the Lastfm dataset). The disagreement is generally much higher in the more complex rank-based measures (0–83.33%) compared to simpler set-based Rel measures (0–50.00%). Therefore, there are many cases where the mean of Fair and Rel scores is not the best case, especially for Lastfm and Jester where Rel scores are higher and vary more. In these two datasets, more often than not, DPFR leads to different conclusions than a simple average. Yet, sometimes the average agrees with DPFR in the best model: for QK-video, disagreement is low (8.33%), and there is a perfect agreement on the best model for Amazon-lb; we posit that these are due to equally poor and low variance in the Rel scores. This is in line with our correlation analysis. As there is a huge range of variability across datasets (0–58.33%), we do not recommend using a simple average to get the same result as DPFR, as it is unreliable and inconsistent. However, in the average is almost always better than existing joint measures. Generally, averaging fails to reach the same conclusion as DPFR almost half the time, especially when there is high variability across the Rel and Fair scores.

## 5.5  Efficient DPFR

*5.5.1  The efficiency of the PF generation.* We study the efficiency of DPFR by comparing the PF, an estimated version of PF on a subset of points, and the Fair+Rel measures. The estimated version of PF uses 3–12 points as per §3.4. We compute the amount of points in the estimated PF as % of those in the PF, and the resulting computation times. One point in the PF translates to one round of computing all Fair and Rel measures, so fewer points means faster. For brevity, we report the estimated PF with only 3, 6, and 12 points in Tab. 3 (see App. D, Tab. 9 for results with other number of points).

The PF (Fig. 2) has hundreds to tens of thousands of points each, while the estimated PF only contains 0.02–2.40% of the points, which means reduced computational complexity for the PF generation. In terms of actual computation time (Tab. 3), computing the PF with Oracle2Fair take 0.56–75.77 mins to compute, but only 0.19–4.17 mins for the PFs estimated with 3 points for all datasets except Jester. For Jester, it takes ~14 hours and the estimation only takes ~9 hours. However, this is expected for Jester as it has 62K users in the test split, as opposed to the 3.5K or fewer in the other datasets (i.e., see Tab. 4 in App. A). While computing the estimated PFs is on

**Table 3: Efficiency and effectiveness comparison between PF, estimated PF (Est. PF), and Fair+Rel measures: percentage of data points in the Est. PF (% pts), computation time. The average distance between midpoints in the Est. PF and PF over 12 measure pairs is denoted as Dist. Minimum agreement (Min τ) is the Kendall's τ correlation between DPFR with PF and Est. PF. Both PF and Est. PF compute 11 Rel and Fair measures simultaneously. The times for other evaluation measures are averaged (Avg/model) and summed (All models) over 16 model combinations.**

| | | Lastfm | Amazon-lb | QK-video | Jester | ML-10M | ML-20M |
|---|---|---|---|---|---|---|---|
| #pts | PF | 4882 | 847 | 499 | 16202 | 2781 | 3783 |
| %pts | Est. PF (12 pts) | 0.25 | 1.42 | 2.40 | 0.07 | 0.43 | 0.32 |
| | Est. PF (6 pts) | 0.12 | 0.71 | 1.20 | 0.04 | 0.22 | 0.16 |
| | Est. PF (3 pts) | 0.06 | 0.35 | 0.60 | 0.02 | 0.11 | 0.08 |
| ↓ Computation time (mins.) | PF | 19.18 | 0.56 | 10.49 | 847.42 | 28.99 | 75.77 |
| | Est. PF (12 pts) | 2.02 | 0.19 | 4.23 | 552.16 | 1.90 | 2.60 |
| | Est. PF (6 pts) | 2.00 | 0.19 | 4.12 | 551.72 | 1.84 | 2.54 |
| | Est. PF (3 pts) | 2.01 | 0.19 | 4.07 | 552.26 | 1.82 | 2.52 |
| Avg/model — IBO | | <0.3s | <0.3s | 0.01 | 0.01 | <0.3s | 0.01 |
| Avg/model — MME | | 2.04 | 0.03 | 19.51 | 0.09 | 15.25 | 89.13 |
| Avg/model — IAA | | <0.3s | <0.3s | 0.01 | 0.02 | 0.01 | 0.02 |
| Avg/model — II-F | | <0.3s | <0.3s | 0.01 | 0.01 | 0.01 | 0.02 |
| Avg/model — AI-F | | <0.3s | <0.3s | 0.01 | 0.01 | <0.3s | 0.01 |
| All models — IBO | | 0.02 | <0.3s | 0.10 | 0.12 | 0.06 | 0.15 |
| All models — MME | | 32.63 | 0.49 | 312.14 | 1.38 | 244.03 | 1426.10 |
| All models — IAA | | 0.03 | <0.3s | 0.10 | 0.36 | 0.13 | 0.30 |
| All models — II-F | | 0.04 | <0.3s | 0.16 | 0.14 | 0.1 | 0.25 |
| All models — AI-F | | 0.03 | <0.3s | 0.11 | 0.13 | 0.07 | 0.17 |
| ↑ Min τ | Est. PF (12 pts) | 0.95 | 1.00 | 1.00 | 0.98 | 0.98 | 0.97 |
| | Est. PF (6 pts) | 0.90 | 0.97 | 1.00 | 0.98 | 0.95 | 0.92 |
| | Est. PF (3 pts) | 0.78 | 0.98 | 1.00 | 1.00 | 0.97 | 0.75 |
| ↓ Dist. | Est. PF (12 pts) | 0.01 | 0.02 | 0.00 | 0.00 | 0.01 | 0.01 |
| | Est. PF (6 pts) | 0.03 | 0.05 | 0.00 | 0.00 | 0.02 | 0.02 |
| | Est. PF (3 pts) | 0.03 | 0.05 | 0.00 | 0.00 | 0.03 | 0.05 |

average slower than computing the joint measures IBO, IAA, II-F, and AI-F, it is expected as the (estimated) PFs compute 11 measures simultaneously. Yet, in most cases (except Amazon-lb and Jester), the estimated PFs is still faster to compute than the time to compute MME for one model per dataset, let alone to compute MME for all models. For ML-20M, computing the estimated PF is even up to 35 times faster than computing MME of one model.

*5.5.2 The effectiveness of efficient DPFR.* We study to what extent the DPFR from the efficiently generated PF (estimated PF) is a reasonable proxy for fairness-relevance joint evaluation using the PF, in terms of giving a similar ordering of models. We compare the DPFR from the PF and estimated PF using Kendall's τ correlations. Further, as DPFR is computed based on a $\alpha$-based reference point lying on the PF, to quantify possible accuracy loss of the estimated PF, in Tab. 3 we also report the error of the midpoint estimation. This error is computed as the Euclidean distance between the reference point in the PF and estimated PF (i.e., the midpoint in our case), following the idea from [45].

We first analyse the error of the midpoint estimation. Across the 12 measure pairs and 6 datasets, the midpoint coordinates on average do not move much, i.e., the distance is 0.00−0.05, even when the PF is only estimated with 3 points. Ergo, the correlations between the rank ordering of models given by the DPFR of PF and its estimation, are still equivalent ($\tau \geq 0.9$) when estimated with 6 or 12 points [23, 44]. Even the 3-point estimation maintains high agreement ($\tau \in [0.75, 1]$), with only 5 cases having $\tau < 0.9$ across

6 datasets and 12 measure-pairs. Therefore, it is possible to only compute a small number of points in the PF, e.g., 6 or 12 points, and still make a reliable PF estimation, evidenced by the small shift of the PF midpoint and the rank ordering of the models remaining equivalent ($\tau \geq 0.9$), if not identical ($\tau = 1$) for all measure pairs and datasets.

# 6 DISCUSSION AND CONCLUSIONS

Recommendation system (RS) evaluation has long been based on measures that quantify only relevance, e.g., NDCG, AP. Recently, the focus of evaluation has shifted to include fairness, for instance measured as the equal opportunity of items to be exposed to users. However, there exists no de-facto, robust approach that can consistently quantify these two aspects. We have proposed a novel approach (DPFR) that incorporates fairness and relevance measures under a joint evaluation scheme for RSs.

DPFR can compute the empirical best possible recommendation, jointly accounting for a given pair of relevance and fairness measures, in a principled way according to Pareto-optimality. DPFR is modular, tractable, and intuitively understandable. It can be used with several existing measures for relevance and fairness, and in principle allows different trade-offs of relevance and fairness to be incorporated into the measurement. We empirically show that existing evaluation measures of fairness w.r.t. relevance [3, 4, 8, 41, 48] behave inconsistently: they disagree with optimal solutions based on DPFR computed on more robust and well understood measures of relevance, such as NDCG, and fairness, such as Gini. We uncover some weaknesses of these measures, but more research is warranted to properly study their behaviour. Admittedly, the existing joint measures are not originally defined to be aligned with existing relevance and fairness measures [11, 14, 24, 25, 34, 56]. Therefore, it is not surprising that they have different results from DPFR. However, existing measures show varying performance also from each other and from well-understood relevance and fairness measures. Thus, DPFR can provide a viable alternative for robust, interpretable, and provenly optimal evaluation strategy in offline scenarios. We also show that DPFR can be computed fast while reaching equivalent conclusion. Overall, DPFR demonstrates distinct benefits in mitigating false conclusions by up to 50% compared to basic aggregation methods like averaging. Surprisingly, simple averaging aligns more with our Pareto-optimal based DPFR, than existing joint measures. We recommend combining either Ent-MAP or Ent-NDCG, as often, the conclusions are distinguishable from simply averaging, or taking the best model based on fairness or relevance measures.

Our experiments are conducted with a wide range of fairness and relevance metrics across several datasets. Nonetheless, it is still possible that there may be other metrics for which our approach is not suitable. For instance, using relevance measures like hit rate or MRR, which rely on single-item relevance, may necessitate adjustments to our approach. Further, our experiments were carried out with a balanced tradeoff between relevance and fairness, a setup that may not align with the evaluation requirements of all scenarios. It is easy to anticipate situations where either fairness or relevance might warrant a greater emphasis. These scenarios require further experimentation. In the future, by modifying the algorithm for PF generation, DPFR could also be extended for other fairness types.

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

## A EXTENDED DATASET STATISTICS

Tab. 4 presents the statistics of each dataset split. For several datasets (e.g., Amazon-lb and ML-*), the number of users in the test split is significantly less than the number of users in the train split. Tab. 5 presents the statistics of items in the test split, per user.

**Table 4: Number of [users, items, and interactions] in the train, validation, and test split after preprocessing.**

|  | train | val | test |
|---|---|---|---|
| Lastfm | [1842, 2821, 42758] | [1831, 2448, 14248] | [1836, 2476, 14237] |
| Amazon-lb | [1054, 552, 8860] | [470, 204, 1811] | [437, 209, 1726] |
| QK-video | [4656, 6245, 34345] | [3470, 4095, 8726] | [3514, 4101, 8706] |
| Jester | [63724, 100, 1294511] | [62137, 100, 427623] | [62167, 100, 427926] |
| ML-10M | [49378, 6838, 4944064] | [2695, 7828, 296914] | [1523, 7880, 121707] |
| ML-20M | [89917, 8719, 9882504] | [4987, 10742, 472243] | [2178, 13935, 233394] |

**Table 5: Statistics of items in the test split, per user, i.e., the number of relevant items per user**

|  | mean | min | median | max |
|---|---|---|---|---|
| Lastfm | 7.75 | 1 | 8 | 19 |
| Amazon-lb | 3.95 | 1 | 3 | 16 |
| QK-video | 2.48 | 1 | 2 | 16 |
| Jester | 6.88 | 1 | 6 | 29 |
| ML-10M | 79.91 | 1 | 46 | 1632 |
| ML-20M | 107.16 | 1 | 53 | 2266 |

## B ALGORITHMS FOR GENERATING PARETO FRONTIER

We present the pseudocodes of the algorithms for generating the Pareto Frontier: the Oracle (Algorithm 1) and ORACLE2FAIR (Algorithm 2).

## C MODIFICATIONS TO THE GS ALGORITHM

The original GS algorithm [46] increases individual item fairness within clusters of similar items. The item similarity is determined based on the item embedding. As our experiments and the FAIR measures do not deal with the additional constraint of item similarity, we consider all items as similar. Therefore, we only have a single cluster of items.

On top of that, we also modify GS to increase computational efficiency. In the original GS algorithm, for each pair of candidate items for replacement $i$ and candidate items to be replaced $i'$, the algorithm finds all users that have $i$ in the original recommendation list. The algorithm then computes the loss in relevance (computed using predicted relevance value) if item $i$ is replaced by $i'$. Until this point, our modified algorithm does the same. The difference is that we save each $i, i', u$, and the loss associated, while the original algorithm only saves the information for the one user $u^*$, whose recommendation list will suffer the least loss when we replace $i$ with $i'$. The original GS then proceeds to make the replacement, update the pool of candidate items for replacement and to be replaced, and go through the entire process again. Initially, we found that with the GS algorithm, around 20% of the initial recommendations are replaced during the process, meaning that for Amazon-lb, there are at least $437 \times 10 \times 0.2 \geq 800$ iterations of the process (Tab. 4). The number of iterations is much bigger for ML-10M, which has more than three times the number of recommendation slots as Amazon-lb, and therefore it is extremely costly to use the GS algorithm as is.

Our modified GS utilises the saved information earlier. After going through all pairs of $(i, i')$, we sort the saved list from the smallest to the largest loss, and (attempt to) perform the replacement using the first $P$ pairs, where $P$ is 25% of the number of recommendation slots. During the replacement process, if the item that is supposed to be replaced no longer exists in the user's recommendation list, we simply skip the replacement.

## D EXTENDED RESULTS

We present the actual scores of the recommender models in Tab. 6–7. In Tab. 8, we present the gradient values of the PF, used in determining which pair of measures are suitable for DPFR. In Fig. 4 we present the Pareto Frontier (PF) of fairness and relevance together with recommender model scores in Tab. 6–7 for Amazon-lb, Jester, and ML-*. In Tab. 9 we present the Kendall's $\tau$ correlation scores of the DPFR from estimated PF and the PF.

---

**Algorithm 1:** Oracle

Create recommendations with the highest relevance

---

**Data:**

$I$: all items in the dataset;

$H_u$: items in train-val split for each user $u \in U$;

$R_u^*$: items in test split (relevant items) for each user $u \in U$;

$k$: number of recommended items

**Result:**

$rec$: most relevant recommendation

$result$: a list of relevance and fairness scores

$itemNotInRec$: items that are not in the recommendation

   /* Handle users with exactly $|R_u^*| = k$                                              */

1 **foreach** $u \in U$ *where* $|R_u^*| = k$ **do** $rec[u] \leftarrow R_u^*$;

   /* Handle users with $|R_u^*| > k$                                                 */

2 **for** $K = k + 1$ **to** $max(|R_u^*|)$ **do**

3     $userWithK \leftarrow$ get users where $|R_u^*| = K$

4     **foreach** $u \in userWithK$ **do**

5         $takenItem[u] \leftarrow R_u^* \setminus rec$

6         $weight[u] \leftarrow sum(countInRec(takenItem[u]))$

7     **end**

8     $sortUserWithK \leftarrow$ sort $userWithK$ by the least weight

9     $tempRec[u] \leftarrow R_u^* \setminus takenItem[u]$

10     keep only max $k$ items in $tempRec[u]$

11     **foreach** $u \in sortUserWithK$ **do**

12         $rec[u] \leftarrow tempRec[u]$

13         $numItemToAdd \leftarrow k - |tempRec[u]|$

14         sort $takenItem[u]$ by the least item count

15         $rec[u].append(takenItem[u][: numItemToAdd])$

16     **end**

17 **end**

   /* Handle users with $|R_u^*| < k$                                                 */

18 $remainUser \leftarrow$ get users where $|R_u^*| < k$

19 **foreach** $u \in remainUser$ **do** $rec[u] \leftarrow |R_u^*|$;

20 $itemNotInRec \leftarrow I \setminus rec$

21 **foreach** $u \in remainUser$ **do**

22     **while** $|rec[u]| < k$ *and* $itemNotInRec \neq \emptyset$ **do**

23         **for** $item \in itemNotInRec$ **do**

24             **if** $item \notin H_u$ **then**

25                 $rec[u].append(item)$

26                 $itemNotInRec \leftarrow itemNotInRec \setminus \{item\}$

27             **end**

28         **end**

29     **end**

30     **if** $\exists u$ *where* $|rec[u]| < k$ **then**

31         **while** $|rec[u]| < k$ **do**

32             $candItem \leftarrow$ least popular item in $rec$ that is not in $H_u \cup R_u^* \cup rec[u]$

33             $rec[u].append(candItem)$

34         **end**

35     **end**

36 **end**

37 $result \leftarrow calculateScores(rec)$

---

---

**Algorithm 2:** ORACLE2FAIR

After recommending maximally relevant items, iteratively change the recommendation list to increase fairness until maximum fairness is reached

---

**Data:** $H_u, R_u^*, I, k$

**Result:**

$rec$: most fair possible recommendation;

$result$: a list of relevance and fairness scores

1   $rec, result, itemNotInRec \leftarrow Oracle(I, H_u, R_u^*)$

    /* Get the most popular item in the recommendations and its frequency count               */

2   $newMostPop \leftarrow mostPop \leftarrow getMostPopItem(rec)$

3   $newCntPop \leftarrow cntPop \leftarrow cnt(mostPop, rec)$

4   $uWithMostPop \leftarrow$ all users with $mostPop \in rec[u]$

5   sort $uWithMostPop$ by largest index of $mostPop$ in $rec[u]$

6   **for** $i \in itemNotInRec$ **do**

7     **if** $cntPop = 1$ **then** break;

8     **if** $newMostPop \neq mostPop$ **then**

9       $mostPop \leftarrow newMostPop$

10      update $uWithMostPop$ following $mostPop$

11     **end**

12    **if** $newMostPop = mostPop$ **then**

13      $candU \leftarrow$ all $u$ in $uWithMostPop$ where $i \notin H_u$

14      **if** $\exists u \in candU$ with $i \in R_u^*$ **then**

15        recommend $i$ to the top $u$ with $i \in R_u^*$

16      **end**

17      **else** recommend $i$ to the top $u$ from $candU$;

18      reorder $rec[u]$ so all relevant items are at the top

19      $result.append(calculateScores(rec))$

20      $itemNotInRec \leftarrow itemNotInRec \setminus \{i\}$

21      $newMostPop \leftarrow getMostPopItem(rec)$

22      $newCntPop \leftarrow cnt(mostPop, rec)$

23     **end**

24   **end**

25   **else**

26     do lines 2–5

27     $i \leftarrow leastPop \leftarrow getLeastPopItem(rec)$

28     $m, n \leftarrow |U|, |I|$

29     **while** $cntPop > \lceil km/n \rceil$ **do**

30      **if** $newMostPop \neq mostPop$ **then** do lines 9–10;

31      **if** $newMostPop = mostPop$ **then**

32        $candU \leftarrow$ all $u$ in $uWithMostPop$ where $i \notin H_u \cup rec[u]$

33        do lines 14–19

34        do lines 26–27

35      **end**

36     **end**

37   **end**

---

**Table 6: Relevance (Rel), fairness (Fair), and joint fairness and relevance (Fair+Rel) scores at $k = 10$ of the recommender models for Lastfm, Amazon-lb, and QK-video, without and with re-ranking the the top $k' = 25$ items using Borda Count (BC), COMBMNZ (CM), and Greedy Substitution (GS) evaluated at $k = 10$. The most relevant or most fair score per measure is in bold. ↑ means the higher the better, ↓ the lower the better.**

| | model | | ItemKNN | | | | BPR | | | | MultiVAE | | | | NCL | | |
|---|---|---|---|---|---|---|---|---|---|---|---|---|---|---|---|---|---|
| | reranking | - | BC | CM | GS | - | BC | CM | GS | - | BC | CM | GS | - | BC | CM | GS |
| **Lastfm** Rel | ↑ HR | 0.765 | 0.742 | 0.581 | 0.750 | 0.773 | 0.729 | 0.587 | 0.751 | 0.778 | 0.693 | 0.523 | 0.734 | **0.793** | 0.726 | 0.571 | 0.765 |
| | ↑ MRR | 0.484 | 0.333 | 0.270 | 0.481 | 0.492 | 0.323 | 0.280 | 0.488 | 0.476 | 0.285 | 0.232 | 0.470 | **0.503** | 0.311 | 0.260 | 0.499 |
| | ↑ P | 0.172 | 0.147 | 0.089 | 0.167 | 0.178 | 0.140 | 0.092 | 0.169 | 0.176 | 0.129 | 0.076 | 0.161 | **0.184** | 0.141 | 0.087 | 0.173 |
| | ↑ MAP | 0.137 | 0.085 | 0.053 | 0.135 | 0.141 | 0.080 | 0.058 | 0.138 | 0.138 | 0.070 | 0.045 | 0.132 | **0.148** | 0.079 | 0.050 | 0.144 |
| | ↑ R | 0.218 | 0.186 | 0.114 | 0.211 | 0.224 | 0.180 | 0.119 | 0.211 | 0.224 | 0.163 | 0.098 | 0.205 | **0.234** | 0.180 | 0.110 | 0.220 |
| | ↑ NDCG | 0.245 | 0.181 | 0.119 | 0.241 | 0.252 | 0.173 | 0.126 | 0.244 | 0.247 | 0.155 | 0.102 | 0.235 | **0.261** | 0.170 | 0.115 | 0.252 |
| Fair | ↑ Jain | 0.042 | 0.101 | 0.094 | 0.046 | 0.058 | 0.151 | 0.140 | 0.067 | 0.097 | **0.236** | 0.222 | 0.115 | 0.082 | 0.216 | 0.215 | 0.095 |
| | ↑ QF | 0.474 | 0.642 | **0.679** | 0.533 | 0.362 | 0.491 | 0.528 | 0.402 | 0.517 | 0.658 | 0.678 | 0.554 | 0.453 | 0.622 | 0.657 | 0.502 |
| | ↑ Ent | 0.589 | 0.727 | 0.735 | 0.622 | 0.610 | 0.736 | 0.740 | 0.646 | 0.707 | 0.820 | **0.826** | 0.740 | 0.671 | 0.801 | 0.810 | 0.705 |
| | ↑ FSat | 0.129 | 0.197 | 0.216 | 0.152 | 0.147 | 0.211 | 0.228 | 0.177 | 0.202 | 0.293 | **0.321** | 0.249 | 0.178 | 0.269 | 0.286 | 0.221 |
| | ↓ Gini | 0.904 | 0.810 | 0.790 | 0.879 | 0.910 | 0.827 | 0.818 | 0.887 | 0.839 | 0.715 | **0.696** | 0.803 | 0.872 | 0.748 | 0.728 | 0.840 |
| Fair+Rel | ↑ IBO | 0.209 | 0.270 | 0.256 | 0.227 | 0.208 | 0.263 | 0.253 | 0.228 | 0.261 | **0.314** | 0.278 | 0.281 | 0.242 | 0.308 | 0.292 | 0.265 |
| | ↓ MME | 0.001 | 0.001 | 0.001 | 0.001 | 0.001 | 0.001 | 0.001 | 0.001 | 0.001 | 0.000 | 0.001 | 0.001 | 0.001 | **0.000** | 0.001 | 0.001 |
| | ↓ IAA | 0.004 | 0.004 | 0.004 | 0.004 | 0.004 | 0.004 | 0.004 | 0.004 | 0.004 | 0.004 | 0.004 | 0.004 | **0.004** | 0.004 | 0.004 | 0.004 |
| | ↓ II-F | 0.001 | 0.001 | 0.002 | 0.001 | 0.001 | 0.001 | 0.002 | 0.001 | 0.001 | 0.001 | 0.002 | 0.001 | **0.001** | 0.001 | 0.002 | 0.001 |
| | ↓ AI-F | 0.000 | 0.000 | 0.000 | 0.000 | 0.000 | 0.000 | 0.000 | 0.000 | 0.000 | 0.000 | 0.000 | 0.000 | 0.000 | **0.000** | 0.000 | 0.000 |
| **Amazon-lb** Rel | ↑ HR | **0.046** | 0.021 | 0.016 | 0.043 | 0.011 | 0.014 | 0.021 | 0.011 | 0.039 | 0.007 | 0.014 | **0.046** | 0.034 | 0.021 | 0.011 | 0.034 |
| | ↑ MRR | 0.020 | 0.011 | 0.011 | 0.020 | 0.003 | 0.005 | 0.007 | 0.003 | 0.023 | 0.003 | 0.004 | **0.024** | 0.022 | 0.006 | 0.003 | 0.022 |
| | ↑ P | 0.005 | 0.002 | 0.002 | 0.005 | 0.001 | 0.001 | 0.002 | 0.001 | 0.004 | 0.001 | 0.002 | **0.005** | 0.004 | 0.002 | 0.001 | 0.004 |
| | ↑ MAP | 0.006 | 0.004 | 0.004 | 0.006 | 0.002 | 0.003 | 0.004 | 0.002 | 0.006 | 0.002 | 0.003 | 0.006 | **0.006** | 0.002 | 0.001 | 0.006 |
| | ↑ R | **0.013** | 0.007 | 0.005 | 0.013 | 0.005 | 0.008 | 0.010 | 0.005 | 0.010 | 0.005 | 0.008 | 0.010 | 0.012 | 0.007 | 0.003 | 0.011 |
| | ↑ NDCG | 0.011 | 0.006 | 0.005 | 0.011 | 0.003 | 0.005 | 0.006 | 0.003 | 0.010 | 0.003 | 0.004 | **0.011** | 0.011 | 0.004 | 0.002 | 0.011 |
| Fair | ↑ Jain | 0.271 | **0.547** | 0.431 | 0.324 | 0.223 | 0.432 | 0.359 | 0.259 | 0.035 | 0.123 | 0.097 | 0.043 | 0.026 | 0.098 | 0.080 | 0.031 |
| | ↑ QF | 0.650 | **0.679** | 0.612 | 0.663 | 0.549 | 0.630 | 0.594 | 0.571 | 0.222 | 0.294 | 0.286 | 0.254 | 0.229 | 0.315 | 0.310 | 0.265 |
| | ↑ Ent | 0.802 | **0.882** | 0.839 | 0.829 | 0.747 | 0.839 | 0.809 | 0.776 | 0.418 | 0.587 | 0.558 | 0.469 | 0.371 | 0.560 | 0.534 | 0.426 |
| | ↑ FSat | 0.370 | **0.538** | 0.438 | 0.435 | 0.314 | 0.410 | 0.376 | 0.364 | 0.114 | 0.159 | 0.152 | 0.138 | 0.091 | 0.146 | 0.138 | 0.115 |
| | ↓ Gini | 0.665 | **0.492** | 0.598 | 0.613 | 0.747 | 0.601 | 0.660 | 0.703 | 0.949 | 0.882 | 0.899 | 0.930 | 0.959 | 0.898 | 0.910 | 0.943 |
| Fair+Rel | ↑ IBO | 0.062 | 0.038 | 0.029 | **0.067** | 0.019 | 0.029 | 0.038 | 0.019 | 0.029 | 0.019 | 0.029 | 0.033 | 0.038 | 0.033 | 0.024 | 0.033 |
| | ↓ MME | 0.001 | 0.001 | 0.001 | 0.001 | 0.001 | **0.001** | 0.001 | 0.001 | 0.003 | 0.001 | 0.001 | 0.003 | 0.004 | 0.001 | 0.001 | 0.004 |
| | ↓ IAA | 0.011 | 0.011 | 0.011 | **0.011** | 0.011 | 0.011 | 0.011 | 0.011 | 0.011 | 0.011 | 0.011 | 0.011 | 0.011 | 0.011 | 0.011 | 0.011 |
| | ↓ II-F | 0.006 | 0.006 | 0.006 | **0.006** | 0.006 | 0.006 | 0.006 | 0.006 | 0.006 | 0.006 | 0.006 | 0.006 | 0.006 | 0.006 | 0.006 | 0.006 |
| | ↓ AI-F | 0.000 | **0.000** | 0.000 | 0.000 | 0.000 | 0.000 | 0.000 | 0.000 | 0.001 | 0.000 | 0.000 | 0.001 | 0.002 | 0.000 | 0.000 | 0.002 |
| **QK-video** Rel | ↑ HR | 0.040 | 0.046 | 0.047 | 0.038 | 0.099 | 0.063 | 0.045 | 0.089 | 0.109 | 0.089 | 0.061 | 0.103 | **0.130** | 0.102 | 0.077 | 0.124 |
| | ↑ MRR | 0.013 | 0.014 | 0.013 | 0.013 | 0.039 | 0.018 | 0.015 | 0.038 | 0.039 | 0.028 | 0.021 | 0.038 | **0.048** | 0.030 | 0.024 | 0.047 |
| | ↑ P | 0.004 | 0.005 | 0.005 | 0.004 | 0.011 | 0.007 | 0.005 | 0.010 | 0.012 | 0.009 | 0.006 | 0.011 | **0.014** | 0.011 | 0.008 | 0.013 |
| | ↑ MAP | 0.005 | 0.005 | 0.005 | 0.005 | 0.017 | 0.008 | 0.006 | 0.016 | 0.018 | 0.012 | 0.009 | 0.017 | **0.022** | 0.013 | 0.010 | 0.021 |
| | ↑ R | 0.014 | 0.018 | 0.019 | 0.014 | 0.043 | 0.028 | 0.019 | 0.039 | 0.051 | 0.039 | 0.027 | 0.047 | **0.061** | 0.045 | 0.033 | 0.058 |
| | ↑ NDCG | 0.009 | 0.011 | 0.010 | 0.009 | 0.029 | 0.015 | 0.011 | 0.027 | 0.031 | 0.022 | 0.016 | 0.030 | **0.038** | 0.025 | 0.019 | 0.037 |
| Fair | ↑ Jain | 0.483 | **0.815** | 0.589 | 0.567 | 0.081 | 0.333 | 0.379 | 0.101 | 0.012 | 0.038 | 0.032 | 0.014 | 0.020 | 0.076 | 0.071 | 0.023 |
| | ↑ QF | 0.901 | **0.956** | 0.790 | 0.924 | 0.625 | 0.809 | 0.823 | 0.678 | 0.100 | 0.155 | 0.163 | 0.127 | 0.201 | 0.331 | 0.365 | 0.253 |
| | ↑ Ent | 0.933 | **0.979** | 0.937 | 0.950 | 0.755 | 0.888 | 0.903 | 0.792 | 0.420 | 0.557 | 0.547 | 0.458 | 0.507 | 0.667 | 0.674 | 0.549 |
| | ↑ FSat | 0.443 | **0.659** | 0.547 | 0.522 | 0.212 | 0.346 | 0.382 | 0.259 | 0.052 | 0.089 | 0.090 | 0.070 | 0.077 | 0.140 | 0.150 | 0.104 |
| | ↓ Gini | 0.472 | **0.235** | 0.442 | 0.397 | 0.807 | 0.613 | 0.570 | 0.761 | 0.982 | 0.957 | 0.959 | 0.976 | 0.966 | 0.909 | 0.902 | 0.952 |
| Fair+Rel | ↑ IBO | 0.033 | 0.038 | 0.038 | 0.035 | 0.054 | 0.050 | 0.036 | 0.052 | 0.031 | 0.042 | 0.036 | 0.033 | 0.043 | **0.060** | 0.054 | 0.047 |
| | ↓ MME | 0.000 | **0.000** | 0.000 | 0.000 | 0.000 | 0.000 | 0.000 | 0.000 | 0.000 | 0.000 | 0.000 | 0.000 | 0.000 | 0.000 | 0.000 | 0.000 |
| | ↓ IAA | 0.001 | 0.001 | 0.001 | 0.001 | 0.001 | 0.001 | 0.001 | 0.001 | 0.001 | 0.001 | 0.001 | 0.001 | **0.001** | 0.001 | 0.001 | 0.001 |
| | ↓ II-F | 0.001 | 0.001 | 0.001 | 0.001 | 0.001 | 0.001 | 0.001 | 0.001 | 0.001 | 0.001 | 0.001 | 0.001 | **0.001** | 0.001 | 0.001 | 0.001 |
| | ↓ AI-F | 0.000 | **0.000** | 0.000 | 0.000 | 0.000 | 0.000 | 0.000 | 0.000 | 0.000 | 0.000 | 0.000 | 0.000 | 0.000 | 0.000 | 0.000 | 0.000 |

**Table 7: Relevance (Rel), fairness (Fair), and joint fairness and relevance (Fair+Rel) scores at $k = 10$ of the recommender models for Jester and ML-*, without and with re-ranking the the top $k' = 25$ items using Borda Count (BC), COMBMNZ (CM), and Greedy Substitution (GS) evaluated at $k = 10$. The most relevant or most fair score per measure is in bold. ↑ means the higher the better, ↓ the lower the better.**

| | | model | ItemKNN | | | | BPR | | | | MultiVAE | | | | NCL | | | |
|---|---|---|---|---|---|---|---|---|---|---|---|---|---|---|---|---|---|---|
| | | reranking | - | BC | CM | GS | - | BC | CM | GS | - | BC | CM | GS | - | BC | CM | GS |
| Jester | Rel | ↑ HR | 0.933 | 0.888 | 0.652 | 0.932 | 0.929 | 0.876 | 0.742 | 0.928 | **0.944** | 0.899 | 0.818 | 0.944 | 0.939 | 0.893 | 0.804 | 0.939 |
| | | ↑ MRR | 0.632 | 0.443 | 0.307 | 0.632 | 0.635 | 0.455 | 0.322 | 0.635 | **0.661** | 0.465 | 0.370 | 0.661 | 0.651 | 0.479 | 0.349 | 0.651 |
| | | ↑ P | 0.334 | 0.250 | 0.144 | 0.333 | 0.330 | 0.243 | 0.163 | 0.329 | **0.351** | 0.262 | 0.194 | 0.351 | 0.342 | 0.257 | 0.185 | 0.341 |
| | | ↑ MAP | 0.352 | 0.198 | 0.101 | 0.352 | 0.348 | 0.195 | 0.112 | 0.348 | **0.379** | 0.208 | 0.145 | 0.379 | 0.367 | 0.211 | 0.133 | 0.367 |
| | | ↑ R | 0.529 | 0.393 | 0.197 | 0.529 | 0.524 | 0.377 | 0.255 | 0.523 | **0.555** | 0.405 | 0.324 | 0.555 | 0.543 | 0.400 | 0.305 | 0.542 |
| | | ↑ NDCG | 0.496 | 0.336 | 0.189 | 0.496 | 0.493 | 0.331 | 0.216 | 0.492 | **0.525** | 0.350 | 0.265 | 0.524 | 0.512 | 0.352 | 0.249 | 0.511 |
| | Fair | ↑ Jain | 0.343 | 0.556 | 0.445 | 0.345 | 0.377 | **0.583** | 0.547 | 0.380 | 0.295 | 0.544 | 0.509 | 0.297 | 0.351 | 0.504 | 0.534 | 0.354 |
| | | ↑ QF* | **1.000** | **1.000** | **1.000** | **1.000** | **1.000** | **1.000** | **1.000** | **1.000** | 0.967 | **1.000** | **1.000** | **1.000** | **1.000** | **1.000** | **1.000** | **1.000** |
| | | ↑ Ent | 0.702 | 0.854 | 0.784 | 0.705 | 0.754 | **0.875** | 0.857 | 0.757 | 0.648 | 0.852 | 0.839 | 0.651 | 0.722 | 0.838 | 0.855 | 0.725 |
| | | ↑ FSat | 0.267 | **0.378** | 0.289 | 0.267 | 0.244 | 0.344 | 0.333 | 0.244 | 0.256 | 0.344 | 0.300 | 0.256 | 0.222 | 0.344 | 0.311 | 0.222 |
| | | ↓ Gini | 0.687 | 0.502 | 0.595 | 0.685 | 0.632 | **0.467** | 0.495 | 0.629 | 0.738 | 0.506 | 0.520 | 0.735 | 0.668 | 0.528 | 0.502 | 0.665 |
| | Fair+Rel | ↑ IBO | 0.600 | **0.930** | 0.740 | 0.600 | 0.840 | 0.910 | 0.780 | 0.840 | 0.500 | 0.870 | 0.810 | 0.500 | 0.740 | 0.920 | 0.780 | 0.740 |
| | | ↓ MME | 0.003 | 0.003 | 0.006 | 0.003 | 0.004 | **0.002** | 0.005 | 0.004 | 0.008 | 0.003 | 0.004 | 0.008 | 0.004 | 0.003 | 0.006 | 0.004 |
| | | ↓ IAA | 0.081 | 0.093 | 0.104 | 0.081 | 0.081 | 0.094 | 0.103 | 0.081 | 0.078 | 0.092 | 0.100 | **0.078** | 0.079 | 0.092 | 0.101 | 0.079 |
| | | ↓ II-F | 0.028 | 0.035 | 0.040 | 0.028 | 0.029 | 0.035 | 0.040 | 0.029 | 0.027 | 0.035 | 0.038 | **0.027** | 0.028 | 0.034 | 0.038 | 0.028 |
| | | ↓ AI-F | 0.002 | 0.002 | 0.003 | 0.002 | 0.002 | **0.001** | 0.002 | 0.002 | 0.003 | 0.002 | 0.002 | 0.003 | 0.002 | 0.002 | 0.002 | 0.002 |
| ML-10M | Rel | ↑ HR | 0.487 | 0.480 | 0.443 | 0.481 | 0.512 | 0.462 | 0.386 | 0.485 | 0.417 | 0.438 | 0.387 | 0.410 | **0.521** | 0.473 | 0.402 | 0.513 |
| | | ↑ MRR | 0.282 | 0.242 | 0.225 | 0.279 | 0.299 | 0.208 | 0.185 | 0.295 | 0.237 | 0.231 | 0.191 | 0.235 | **0.302** | 0.216 | 0.203 | 0.301 |
| | | ↑ P | 0.137 | 0.128 | 0.105 | 0.133 | 0.146 | 0.114 | 0.088 | 0.132 | 0.107 | 0.111 | 0.096 | 0.105 | **0.154** | 0.123 | 0.094 | 0.149 |
| | | ↑ MAP | 0.089 | 0.074 | 0.060 | 0.086 | 0.095 | 0.061 | 0.047 | 0.088 | 0.067 | 0.067 | 0.054 | 0.066 | **0.101** | 0.067 | 0.052 | 0.099 |
| | | ↑ R | 0.022 | 0.022 | 0.018 | 0.022 | 0.025 | 0.019 | 0.012 | 0.023 | 0.020 | 0.021 | 0.016 | 0.021 | **0.026** | 0.020 | 0.013 | 0.026 |
| | | ↑ NDCG | 0.150 | 0.133 | 0.113 | 0.147 | 0.160 | 0.115 | 0.092 | 0.150 | 0.119 | 0.121 | 0.100 | 0.118 | **0.167** | 0.123 | 0.100 | 0.164 |
| | Fair | ↑ Jain | 0.011 | 0.026 | 0.027 | 0.012 | 0.037 | 0.100 | **0.115** | 0.044 | 0.003 | 0.005 | 0.006 | 0.004 | 0.024 | 0.063 | 0.069 | 0.027 |
| | | ↑ QF | 0.044 | 0.062 | 0.068 | 0.047 | 0.145 | 0.199 | **0.216** | 0.160 | 0.014 | 0.021 | 0.025 | 0.016 | 0.086 | 0.123 | 0.132 | 0.094 |
| | | ↑ Ent | 0.407 | 0.503 | 0.514 | 0.418 | 0.596 | 0.697 | **0.716** | 0.624 | 0.238 | 0.302 | 0.324 | 0.258 | 0.519 | 0.625 | 0.638 | 0.544 |
| | | ↑ FSat | 0.044 | 0.062 | 0.068 | 0.047 | 0.145 | 0.199 | **0.216** | 0.160 | 0.014 | 0.021 | 0.025 | 0.016 | 0.086 | 0.123 | 0.132 | 0.094 |
| | | ↓ Gini | 0.987 | 0.973 | 0.971 | 0.985 | 0.945 | 0.895 | **0.879** | 0.932 | 0.997 | 0.994 | 0.993 | 0.996 | 0.969 | 0.936 | 0.930 | 0.963 |
| | Fair+Rel | ↑ IBO | 0.031 | 0.043 | 0.046 | 0.034 | 0.069 | 0.089 | **0.091** | 0.076 | 0.012 | 0.016 | 0.018 | 0.014 | 0.054 | 0.073 | 0.074 | 0.058 |
| | | ↓ MME | 0.001 | 0.001 | 0.001 | 0.001 | 0.001 | **0.001** | 0.001 | 0.001 | 0.003 | 0.002 | 0.001 | 0.002 | 0.001 | 0.001 | 0.001 | 0.001 |
| | | ↓ IAA | 0.008 | 0.009 | 0.009 | 0.008 | 0.008 | 0.009 | 0.009 | 0.008 | 0.009 | 0.009 | 0.009 | 0.009 | **0.008** | 0.009 | 0.009 | 0.008 |
| | | ↓ II-F | 0.000 | 0.000 | 0.000 | 0.000 | 0.000 | 0.000 | 0.000 | 0.000 | 0.000 | 0.000 | 0.000 | 0.000 | **0.000** | 0.000 | 0.000 | 0.000 |
| | | ↓ AI-F | 0.000 | 0.000 | 0.000 | 0.000 | 0.000 | **0.000** | 0.000 | 0.000 | 0.000 | 0.000 | 0.000 | 0.000 | 0.000 | 0.000 | 0.000 | 0.000 |
| ML-20M | Rel | ↑ HR | 0.488 | 0.473 | 0.420 | 0.483 | **0.505** | 0.444 | 0.392 | 0.483 | 0.489 | 0.432 | 0.391 | 0.465 | **0.505** | 0.453 | 0.388 | 0.493 |
| | | ↑ MRR | 0.280 | 0.237 | 0.213 | 0.278 | 0.293 | 0.205 | 0.190 | 0.290 | 0.259 | 0.193 | 0.180 | 0.256 | **0.293** | 0.206 | 0.193 | 0.292 |
| | | ↑ P | 0.142 | 0.131 | 0.106 | 0.139 | 0.145 | 0.116 | 0.094 | 0.136 | 0.141 | 0.112 | 0.091 | 0.128 | **0.150** | 0.121 | 0.094 | 0.141 |
| | | ↑ MAP | 0.092 | 0.077 | 0.061 | 0.090 | 0.096 | 0.063 | 0.052 | 0.092 | 0.089 | 0.060 | 0.049 | 0.082 | **0.100** | 0.068 | 0.053 | 0.095 |
| | | ↑ R | 0.019 | 0.017 | 0.014 | 0.019 | 0.019 | 0.014 | 0.012 | 0.018 | 0.019 | 0.014 | 0.011 | 0.018 | **0.020** | 0.016 | 0.011 | 0.020 |
| | | ↑ NDCG | 0.154 | 0.135 | 0.112 | 0.151 | 0.158 | 0.116 | 0.098 | 0.152 | 0.148 | 0.111 | 0.093 | 0.139 | **0.163** | 0.121 | 0.099 | 0.157 |
| | Fair | ↑ Jain | 0.008 | 0.017 | 0.018 | 0.009 | 0.028 | 0.068 | **0.081** | 0.033 | 0.029 | 0.070 | 0.074 | 0.034 | 0.018 | 0.044 | 0.049 | 0.021 |
| | | ↑ QF | 0.035 | 0.047 | 0.051 | 0.037 | 0.114 | 0.154 | **0.165** | 0.125 | 0.117 | 0.146 | 0.154 | 0.126 | 0.074 | 0.103 | 0.112 | 0.082 |
| | | ↑ Ent | 0.399 | 0.483 | 0.491 | 0.411 | 0.581 | 0.670 | **0.690** | 0.606 | 0.591 | 0.669 | 0.680 | 0.615 | 0.517 | 0.608 | 0.624 | 0.541 |
| | | ↑ FSat | 0.035 | 0.047 | 0.051 | 0.037 | 0.114 | 0.154 | **0.165** | 0.125 | 0.117 | 0.146 | 0.154 | 0.126 | 0.074 | 0.103 | 0.112 | 0.082 |
| | | ↓ Gini | 0.991 | 0.982 | 0.981 | 0.990 | 0.960 | 0.926 | **0.914** | 0.951 | 0.957 | 0.927 | 0.920 | 0.948 | 0.976 | 0.953 | 0.947 | 0.971 |
| | Fair+Rel | ↑ IBO | 0.021 | 0.031 | 0.033 | 0.022 | 0.049 | 0.064 | **0.067** | 0.054 | 0.052 | 0.064 | 0.065 | 0.056 | 0.039 | 0.051 | 0.054 | 0.042 |
| | | ↓ MME | 0.001 | 0.001 | 0.001 | 0.001 | 0.001 | 0.000 | **0.000** | 0.001 | 0.001 | 0.000 | 0.000 | 0.001 | 0.001 | 0.001 | 0.001 | 0.001 |
| | | ↓ IAA | 0.007 | 0.007 | 0.007 | 0.007 | 0.007 | 0.007 | 0.007 | 0.007 | 0.007 | 0.007 | 0.007 | 0.007 | **0.007** | 0.007 | 0.007 | 0.007 |
| | | ↓ II-F | 0.000 | 0.000 | 0.000 | 0.000 | 0.000 | 0.000 | 0.000 | 0.000 | 0.000 | 0.000 | 0.000 | 0.000 | **0.000** | 0.000 | 0.000 | 0.000 |
| | | ↓ AI-F | 0.000 | 0.000 | 0.000 | 0.000 | 0.000 | 0.000 | **0.000** | 0.000 | 0.000 | 0.000 | 0.000 | 0.000 | 0.000 | 0.000 | 0.000 | 0.000 |

*QF = 1 means that all items in the dataset appear in the recommendation across all users.

†The scores of QF are the same as FSat for ML-*, as QF is computed based on the percentage of items in the dataset that are recommended, which in this dataset is equivalent to FSat: the percentage of items in the dataset that are recommended at least $\lfloor \frac{km}{n} \rfloor = 1$ time.

**Table 8: The gradient values of the PF, based on the extreme points (starting and ending points). We consider a gradient to be 'good' if it is not zero or undefined (-).**

|          | Lastfm   | Amazon-lb | QK-video | Jester  | ML-10M  | ML-20M | # good | conclusion  |
|----------|----------|-----------|----------|---------|---------|--------|--------|-------------|
| HR-Ent   | -97.57   | -1.86     | -0.31    | -       | -14.74  | -6.95  | 5      | inconsistent |
| HR-FSat  | -1439.17 | -19.92    | 0.00     | -       | -30.48  | -18.97 | 4      | inconsistent |
| HR-Gini  | 561.63   | 6.23      | 3.71     | -       | 117.19  | 43.44  | 5      | inconsistent |
| HR-Jain  | -979.86  | -18.77    | -5.80    | -       | -157.73 | -78.22 | 5      | inconsistent |
| HR-QF    | 0.00     | 0.00      | 0.00     | -       | -30.48  | -18.97 | 2      | inconsistent |
| MAP-Ent  | -0.17    | -0.17     | -0.03    | -0.07   | -0.14   | -0.18  | 6      | always good |
| MAP-FSat | -2.46    | -1.81     | 0.00     | -44.47  | -0.29   | -0.48  | 5      | inconsistent |
| MAP-Gini | 0.96     | 0.56      | 0.34     | 1.42    | 1.12    | 1.10   | 6      | always good |
| MAP-Jain | -1.68    | -1.70     | -0.54    | -0.37   | -1.51   | -1.98  | 6      | always good |
| MAP-QF   | 0.00     | 0.00      | 0.00     | 0.0     | -0.29   | -0.48  | 2      | inconsistent |
| MRR-Ent  | -97.57   | -1.86     | -0.31    | -       | -14.74  | -6.95  | 5      | inconsistent |
| MRR-FSat | -1439.17 | -19.92    | 0.00     | -       | -30.48  | -18.97 | 4      | inconsistent |
| MRR-Gini | 561.63   | 6.23      | 3.71     | -       | 117.19  | 43.44  | 5      | inconsistent |
| MRR-Jain | -979.86  | -18.77    | -5.80    | -       | -157.73 | -78.22 | 5      | inconsistent |
| MRR-QF   | 0.00     | 0.00      | 0.00     | -       | -30.48  | -18.97 | 2      | inconsistent |
| NDCG-Ent | -0.24    | -0.22     | -0.04    | -0.1    | -0.20   | -0.25  | 6      | always good |
| NDCG-FSat| -3.50    | -2.32     | 0.00     | -68.56  | -0.42   | -0.68  | 5      | inconsistent |
| NDCG-Gini| 1.37     | 0.73      | 0.47     | 2.2     | 1.62    | 1.55   | 6      | always good |
| NDCG-Jain| -2.38    | -2.19     | -0.73    | -0.57   | -2.18   | -2.79  | 6      | always good |
| NDCG-QF  | 0.00     | 0.00      | 0.00     | 0.0     | -0.42   | -0.68  | 2      | inconsistent |
| P-Ent    | -0.20    | -0.33     | -0.07    | -0.08   | -0.16   | -0.20  | 6      | always good |
| P-FSat   | -2.95    | -3.53     | 0.00     | -51.41  | -0.33   | -0.55  | 5      | inconsistent |
| P-Gini   | 1.15     | 1.10      | 0.89     | 1.65    | 1.26    | 1.26   | 6      | always good |
| P-Jain   | -2.01    | -3.33     | -1.40    | -0.43   | -1.70   | -2.27  | 6      | always good |
| P-QF     | 0.00     | 0.00      | 0.00     | 0.0     | -0.33   | -0.55  | 2      | inconsistent |
| R-Ent    | -0.17    | -0.17     | -0.03    | -0.07   | -0.26   | -0.30  | 6      | always good |
| R-FSat   | -2.57    | -1.83     | 0.00     | -48.04  | -0.53   | -0.82  | 5      | inconsistent |
| R-Gini   | 1.00     | 0.57      | 0.34     | 1.54    | 2.04    | 1.88   | 6      | always good |
| R-Jain   | -1.75    | -1.73     | -0.54    | -0.4    | -2.75   | -3.39  | 6      | always good |
| R-QF     | 0.00     | 0.00      | 0.00     | 0.0     | -0.53   | -0.82  | 2      | inconsistent |

**Table 9: Range of agreement $\tau$ between estimated PF and PF across 12 measure pairs, using the estimated PF with 3–12 points.**

| #pts | Lastfm      | Amazon-lb   | QK-video    | Jester      | ML-10M      | ML-20M      |
|------|-------------|-------------|-------------|-------------|-------------|-------------|
| 3    | 0.78−1.00   | 0.98−1.00   | 1.00−1.00   | 1.00−1.00   | 0.97−1.00   | 0.75−1.00   |
| 4    | 0.88−1.00   | 0.98−1.00   | 0.98−1.00   | 0.98−1.00   | 0.98−1.00   | 0.93−1.00   |
| 5    | 0.78−1.00   | 0.98−1.00   | 1.00−1.00   | 1.00−1.00   | 0.97−1.00   | 0.92−1.00   |
| 6    | 0.90−1.00   | 0.97−1.00   | 1.00−1.00   | 0.98−1.00   | 0.95−1.00   | 0.92−1.00   |
| 7    | 0.88−1.00   | 1.00−1.00   | 1.00−1.00   | 1.00−1.00   | 0.98−1.00   | 0.93−1.00   |
| 8    | 0.90−1.00   | 0.98−1.00   | 1.00−1.00   | 0.98−1.00   | 1.00−1.00   | 0.95−1.00   |
| 9    | 0.98−1.00   | 1.00−1.00   | 1.00−1.00   | 1.00−1.00   | 0.97−1.00   | 0.98−1.00   |
| 10   | 0.88−1.00   | 1.00−1.00   | 1.00−1.00   | 0.98−1.00   | 1.00−1.00   | 0.95−1.00   |
| 11   | 0.92−1.00   | 1.00−1.00   | 1.00−1.00   | 1.00−1.00   | 0.98−1.00   | 0.97−1.00   |
| 12   | 0.95−1.00   | 1.00−1.00   | 1.00−1.00   | 0.98−1.00   | 0.98−1.00   | 0.97−1.00   |

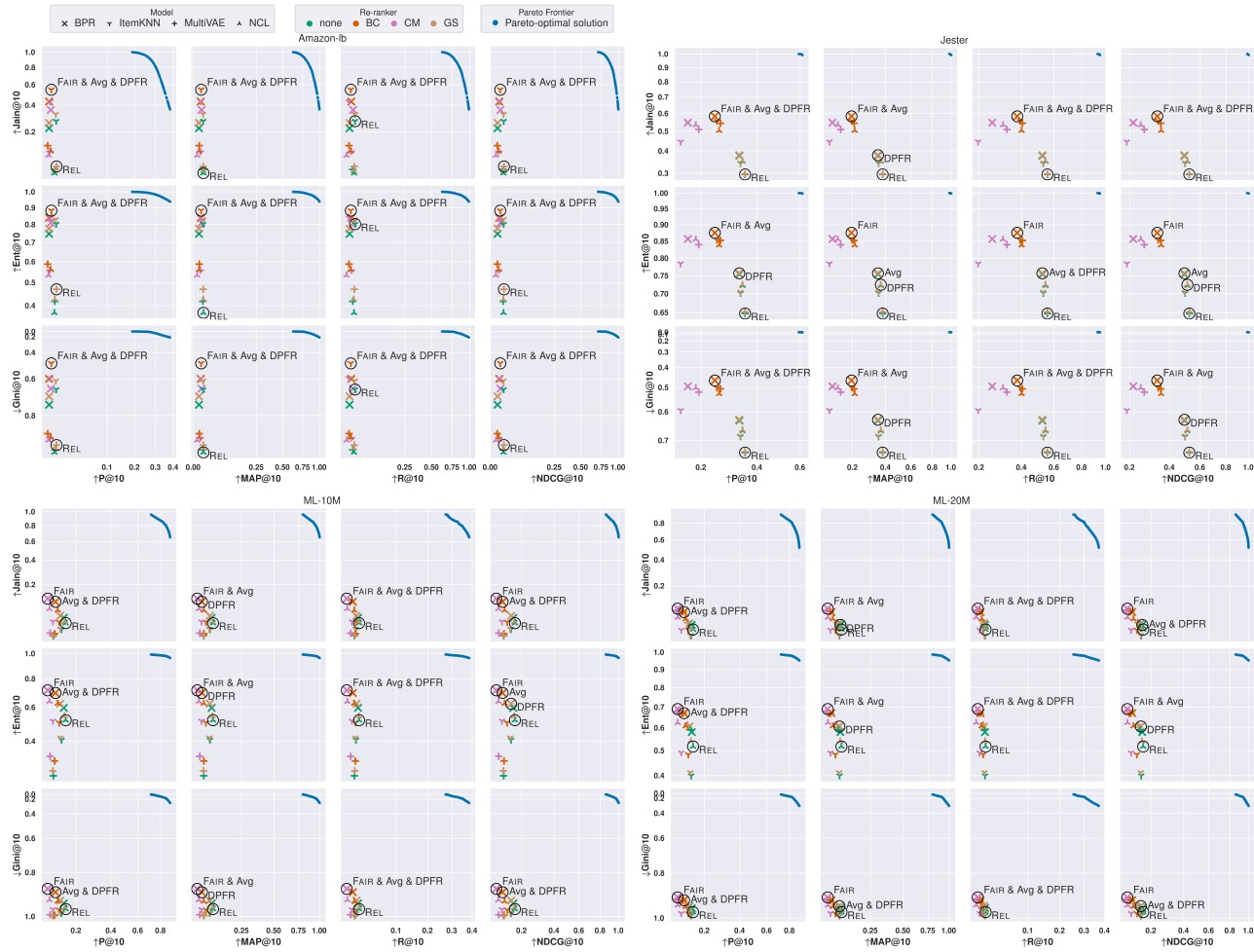

**Figure 4: Pareto Frontier of fairness and relevance (in blue), together with recommender model scores for Amazon-lb, Jester, and ML-\*. FAIR measures are on the $y$-axis and REL measures are on the $x$-axis. We implement exponential-like scales to enhance the visibility of the model plots. The REL, FAIR, Avg, and DPFR denote the best model based on each evaluation approach.**

