# OpenReview forum: "Joint Evaluation of Fairness and Relevance in Recommender Systems with Pareto Frontier"
_ACM.org/TheWebConf/2025/Conference — WWW 2025 Oral_

### Official Review · Reviewer_XYkR · 2024-11-05

**Novelty:** 5
**Technical Quality:** 6

**Review:**

Summary:

Relevance and fairness have always been two important aspects of evaluating recommendation systems, and how to select the "best" recommendation system based on these two criteria has been a challenging problem for many researchers. Some existing work has proposed some combined metrics, but they are mostly ad-hoc, unstable, and do not account very well for both aspects simultaneously. There is also work that evaluates relevance and fairness by using a different measure for each aspect. However, this does not always provide a reliable joint estimate of the goodness of the models, as it may have two different best models. Moreover, due to the different scales of these two metrics, it is difficult to merge them. The authors of this paper, however, take an alternative approach and contribute an approach that builds on the set of all Pareto-optimal solutions. The model selects the best model by computing the distance of the model scores to a desired fairness-relevance balance. The authors have thoroughly demonstrated and detailed the superiority and robustness of the method through extensive experiments and analysis.

Strengths：

1.	The field studied in this paper holds value for promoting the development of the web. Fairness is an important issue in today's recommendation systems, and how to select a more fair model without sacrificing too much relevance of the recommendation system is a difficult problem that plagues many researchers. This paper proposes a concise and modular approach (DPFR) that can effectively solve this challenge.

2.	This paper is the first to use Pareto-optimal solutions to balance relevance metrics and ranking metrics in recommendations, a novel approach that is refreshing and highly innovative.

3.	The paper provides a very detailed description of the experimental setup, which is easy to replicate, and conducts a large number of thorough experiments to demonstrate the superiority and robustness of the proposed method. The results of the experiments are highly convincing.

4.	The writing of this paper is clear, concise, and easy to understand, with a very detailed explanation of the methodology. The appendix also provides the corresponding pseudocode.

Weaknesses：

1.	The conference requirements mentioned that the organizers hope authors will discuss the connection between their work and the web on the first page of their paper, and they wish for authors to supplement this point.

2.	The current authors have not made their experimental code publicly available in the paper, and there is a hope that they will open-source the relevant experimental code.

**Questions:**

Please refer to weaknesses in Review for details.

**Reviewer Confidence:**

3: The reviewer is confident but not certain that the evaluation is correct

**Scope:**

3: The work is somewhat relevant to the Web and to the track, and is of narrow interest to a sub-community

---

### Official Review · Reviewer_fk2C · 2024-11-17

**Novelty:** 4
**Technical Quality:** 4

**Review:**

This paper presents a new approach for jointly evaluating fairness and relevance in recommender systems (RSs) called the Distance from Pareto Frontier (DPFR). Given a user-item interaction dataset, the paper computes the Pareto frontier for a pair of existing relevance and fairness metrics and then uses the distance from this Pareto frontier (PF) as a standard to measure the jointly achievable fairness and relevance.

### Strengths

- **Meaningful Research Focus**: Investigating ways to better evaluate fairness and performance in recommender systems is inherently valuable and significant.
- **Clear Motivation**: The paper provides a very clear motivation for the proposed approach.
- **Versatile Metric Framework**: It introduces a metric computation framework compatible with various existing metrics.

### Weaknesses

- **W1**: **Methodological Unclarities**
  - **Q1-1**: During the replacement process, how is it ensured that each item's maximum recommendation count is $km/n$?
  - **Q1-2**: When replacements are made, the most frequently recommended items are replaced. Since these items are typically the ones most likely to appear in the recommendation lists generated by the model, would replacing them significantly affect the overall recommendation performance? Is such an impact reasonable?
  - **Q1-3**: Some fairness metrics involve the ranking of items. Would the reranking operation during replacements affect these metrics?
  - **Q1-4**: Considering Q1-2 and Q1-3, can the replacement operation guarantee that the new recommendation list still lies on the Pareto frontier? An intuitive and formal explanation from the authors would be appreciated.

- **W2**: **Incomplete Experimental Evaluation**
  - **Q2-1**: In the experiments, does BPR refer to BPR with Matrix Factorization?
  - **Q2-2**: It would be helpful to see the numerical values of the distances to the PF for each method, as Figure 4 does not clearly show significant distance differences.
  - **Q2-3**: Since the ranges of relevance and fairness metrics differ across datasets, is it unreasonable to uniformly set alpha = 0.5?
  - **Q2-4**: The paper does not provide the experimental results for Section 5.2.

- **W3**: **Writing and Presentation Issues**
  - **Q3-1**: Some terms are not explained when first introduced. For example, in line 98, what does "ideal balance" refer to-is it the Pareto optimal point or another trade-off metric? In line 111, "PF" is used without prior declaration that it stands for "Pareto Frontier."
  - **Q3-2**: Section 3.2 lacks illustrative figures, making the method somewhat difficult to understand.
  - **Q3-3**: The paper contains numerous forward and backward references, which hampers readability.

**Questions:**

Beyond the questions listed under the weaknesses (Q1-Q3), I have the following queries:

- **Q4**: In the experiments, Figure 3 shows that 'avg' and 'DPFR' already exhibit a high degree of similarity. Why is it necessary to compute DPFR?
- **Q5**: The experiments set k=10. Since the method requires continuous replacement and reranking of the recommendation list, will the experimental results remain consistent when k is larger?
- **Q6**: Does the computational time of the method increase exponentially as k becomes larger? A complexity analysis might be needed to support this aspect.

**Reviewer Confidence:**

3: The reviewer is confident but not certain that the evaluation is correct

**Scope:**

4: The work is relevant to the Web and to the track, and is of broad interest to the community

---

### Official Review · Reviewer_yVJj · 2024-11-30

**Novelty:** 3
**Technical Quality:** 4

**Review:**

Pros: This paper presents a robust and modular framework, DPFR, to evaluate fairness and relevance in recommender systems. Experiments are conducted comprehensively on diverse datasets, recommendation models, and re-ranking strategies, which enhances the credibility of the findings. This paper leverages well-established optimization concepts (Pareto frontier).

Cons: While the method is novel, it heavily relies on established multi-objective optimization concepts, which may reduce the perception of originality. The focus is limited to individual item fairness, leaving out other dimensions such as group fairness. The full computation of the Pareto frontier can be resource-intensive, which may pose challenges for very large datasets.

**Questions:**

1. Please analyze the computational complexity of the large dataset and provide insights into the expected runtime under extreme conditions.

**Reviewer Confidence:**

2: The reviewer is willing to defend the evaluation, but it is likely that the reviewer did not understand parts of the paper

**Scope:**

3: The work is somewhat relevant to the Web and to the track, and is of narrow interest to a sub-community

---

### Official Review · Reviewer_SMTi · 2024-12-02

**Novelty:** 6
**Technical Quality:** 6

**Review:**

This paper proposes a joint relevance-fairness metric, based on the distance to the Paretto frontier, addressing in particular individual item fairness problem. The authors define this frontier as the curve in a relevance vs fairness space, above which is impossible to increase fairness without sacrificing relevance. A parameter \alpha allows the selection of a point in the curve to be used as reference to measure the distance, allowing the balancing of importance between the two dimensions. Fairness is defined as a form of equal item opportunity (total fairness == same exposure probability).

I did not find any major shortcoming. The metric is, as the authors claim, intuitive, and at least partially aligned with some joint metrics (being impossible to align with all, since they frequently disagree).

In my opinion, the questions around the efficiency of calculating the frontier and the metric are interesting but quite secondary to the contribution. It would be more interesting to have some sort of qualitative analysis, like a case study.

In any case, this paper is a competent and interesting contribution.

**Questions:**

It would be interesting to know the authors' opinion on how this metric can be operationalized for model learning, and what could be expected of a model optimizing for the metric.

**Reviewer Confidence:**

3: The reviewer is confident but not certain that the evaluation is correct

**Scope:**

4: The work is relevant to the Web and to the track, and is of broad interest to the community